# MiRNAs Expression Profiling in Raw264.7 Macrophages after Nfatc1-Knockdown Elucidates Potential Pathways Involved in Osteoclasts Differentiation

**DOI:** 10.3390/biology10111080

**Published:** 2021-10-22

**Authors:** Roberta Russo, Francesca Zito, Nadia Lampiasi

**Affiliations:** Consiglio Nazionale delle Ricerche, Istituto per la Ricerca e l’Innovazione Biomedica, Via Ugo La Malfa 153, 90146 Palermo, Italy; roberta.russo@irib.cnr.it (R.R.); francesca.zito@irib.cnr.it (F.Z.)

**Keywords:** osteoclastogenesis, differential expression, MAPK pathway, NFATc1, PCR arrays, siRNA transfection

## Abstract

**Simple Summary:**

Bone diseases are a worldwide health public problem, and their management has required extensive studies on bone homeostasis. Among all the various mechanisms involved, understanding those underlying osteoclastogenesis is of paramount importance. Here, we tried to elucidate the possible role of miRNAs differentially expressed in RANKL-stimulated RAW264.7 cells expressing the transcription factor NFATc1, the master regulator of osteoclasts generation, and in NFATc1-silenced RAW264.7 cells, which are depleted of NFATc1 by 80%. We performed miRNAs PCR array followed by bioinformatic analysis to discover new possible miRNAs and their targets involved in this process. The results were interesting and suggested that relatively unknown miRNAs (miR-880 and miR-295) control the phosphorylation/dephosphorylation of proteins/transcription factors such as ERK-p38 and NFATc1, by enzymes (DYRKs and DUSPs). In addition, our results confirmed the role of some miRNAs, already known for their involvement in the process of mature osteoclasts formation. This study contributes to a more complete overview of the early stages of osteoclast formation, including cell migration and fusion.

**Abstract:**

Differentiation of macrophages toward osteoclasts is crucial for bone homeostasis but can be detrimental in disease states, including osteoporosis and cancer. Therefore, understanding the osteoclast differentiation process and the underlying regulatory mechanisms may facilitate the identification of new therapeutic targets. Hereby, we tried to reveal new miRNAs potentially involved in the regulation of early steps of osteoclastogenesis, with a particular focus on those possibly correlated with NFATc1 expression, by studying miRNAs profiling. During the first 24 h of osteoclastogenesis, 38 miRNAs were differentially expressed between undifferentiated and RANKL-stimulated RAW264.7 cells, while 10 miRNAs were differentially expressed between RANKL-stimulated cells transfected with negative control or NFATc1-siRNAs. Among others, the expression levels of miR-411, miR-144 and members of miR-29, miR-30, and miR-23 families changed after RANKL stimulation. Moreover, the potential role of miR-124 during osteoclastogenesis was explored by transient cell transfection with anti-miR-124 or miR-124-mimic. Two relatively unknown miRNAs, miR-880-3p and miR-295-3p, were differentially expressed between RANKL-stimulated/wild-type and RANKL-stimulated/NFATc1-silenced cells, suggesting their possible correlation with NFATc1. KEGG enrichment analyses showed that kinase and phosphatase enzymes were among the predicted targets for many of the studied miRNAs. In conclusion, our study provides new data on the potential role and possible targets of new miRNAs during osteoclastogenesis.

## 1. Introduction

Osteoclastogenesis is a tightly regulated process of differentiation starting from mononuclear cells of the monocyte/macrophage cell lineage leading to multinucleated osteoclasts (OCs). Two specific cytokines stimulate pre-OCs to become mature OCs, i.e., the macrophages-colony stimulating factor (M-CSF) and the receptor activator of nuclear factor-B ligand (RANKL). Many transcription factors are activated through the binding of these two cytokines with their appropriate receptors. In particular, in response to RANKL stimulation, nuclear factor-ĸB (NF-ĸB), NFAT-cytoplasmic 1 (NFATc1), and other regulatory factors, such as c-Fos, PU.1 and TRAF6, are activated [1]. NFATc1 is the master regulator of osteoclastogenesis, and its expression and activation are tightly regulated at several levels [2,3]. In undifferentiated cells (not supplied with RANKL), NFATc1 is poorly expressed [4], heavily phosphorylated [1], and located in the cytoplasm [5]. Pre-OCs exposure to RANKL triggers an increase in free intracellular Ca^2+^ levels, which signals calmodulin (CALM), a calcium binding protein, to activate the Ca^2+^-dependent phosphatase Calcineurin (CN). In turn, this enzyme dephosphorylates NFATc1, thereby inducing its nuclear translocation and activation [1]. As a consequence, NFATc1 modulates the expression of its target genes as well as its own, through a process called “autoamplification” [6]. However, nuclear NFATc1 might be dephosphorylated by glycogen synthase kinase-3 (GSK3β) [1], protein kinase A (PKA) [7], and dual-specificity tyrosine-phosphorylation-regulated 1A (DYRK1A) [8,9], followed by its translocation to the cytoplasm. Nevertheless, it is well established that other levels of regulation might further control the stability of the NFATc1 protein.

Numerous miRNAs have been reported as important regulators of OCs differentiation. For example, miR-7b, miR-21, miR-34a, miR-99b, miR-155, miR-223, miR-365, miR-378, and miR-451 are certainly involved in osteoclastogenesis [10,11,12,13,14,15,16]. In particular, overexpression of miR-7b in RAW264.7 cells, a murine macrophage cell line with the capacity to differentiate into OCs after RANKL stimulation, reduces the number of TRAP-positive cells and the formation of multinucleated cells, whereas its inhibition enhances osteoclastogenesis directly targeting DC-STAMP [17]. MiR-21 is upregulated by c-Fos and, in turn, downregulates the expression of programmed cell death 4 (PDCD4), a negative regulator of osteoclastogenesis [18,19,20,21]. The overexpression of miR-155 represses MITF and PU.1, which are crucial transcription factors for OC differentiation [19,20,22]. MiR-223 regulates both osteoblasts and OCs differentiation [23], in the latter by inhibiting nuclear factor I-A (NFIA) expression and M-CSF receptor levels [11,24]. However, in general, little is known about the regulatory mechanisms used by miRNAs in osteoclastogenesis, and very few studies have focused on those correlated to NFATc1. In mouse bone marrow macrophages (BMMs), miR-124 [25] and miR-506-3p [26] have been shown to inhibit osteoclastogenesis by selectively repressing NFATc1, while miR-193-3p has an osteoprotective effect in ovariectomized mice by inhibiting NFATc1 expression [27]. Recently, studies on inflammatory bone resorption showed that NFATc1 is a key upstream regulator for miR-182 induction [28].

The specific aim of this study was to evaluate new miRNAs potentially involved in osteoclastogenesis, with a particular focus on those that were correlated with NFATc1 expression, by profiling miRNAs in undifferentiated, RANKL-stimulated, and NFATc1-knockdown RAW264.7 cells, using PCR arrays and luciferase assays, followed by in silico data analysis.

## 2. Materials and Methods

### 2.1. RAW264.7 Cell Culture

The murine RAW264.7 macrophage cell line was purchased from the American Type Culture Collection (Manassas, VA, USA). Cells were grown in Dulbecco’s Modified Eagle’s Medium (DMEM, Gibco, NY, USA) with 10% heat-inactivated fetal bovine serum (FBS, Sigma Aldrich., St. Louis, MO, USA), 100 U/mL penicillin, and 100 µg/mL streptomycin. To induce osteoclast differentiation, cells were cultured in alpha-minimal essential medium (α-MEM, Gibco Laboratories, Grand Island, NY, USA) with 10% heat-inactivated fetal bovine serum (FBS, Sigma Aldrich., St. Louis, MO, USA), 100 U/mL penicillin, and 100 µg/mL streptomycin with RANKL 50 ng/mL (PeproTech, East Windsor, NJ, USA).

### 2.2. Small Interfering RNA (siRNA) Transfection

Cells were transfected with NFATc1-siRNA or All Star negative control (NC) siRNA (Qiagen, Germantown, MD, USA), as previously reported [4]. All transfections were carried out with 20 nM duplex siRNA in α-MEM medium without FBS or antibiotics. Six hours later, we added RANKL (50 ng/mL) to the medium. One or two days after transfection, cells were harvested to perform further analyses. Experiments were repeated three times.

### 2.3. miRNAs Profiler PCR Array Analysis

Mouse miRNAs Profiler PCR array (miScript miRNA PCR Array Mouse miFinder GeneGlobe ID—MIMM-001Z), in 96-well plate format (Qiagen Sciences, Germantown, MD, USA), was used to analyze miRNAs expression changes. Samples were prepared from RNA extracted from RAW264.7 cells using the RNeasy MinElute Cleanup Kit (Qiagen Sciences, Germantown, MD, USA). RAW264.7 cells were untransfected (−/+RANKL) or transfected in the presence of RANKL (+RANKL) (negative control (NC)-siRNA or NFATc1-siRNA). Total RNA was reverse-transcribed in cDNA with miScript II RT Kit according to Qiagen’s instructions. QPCR was run as previously described [29], with miScript SYBR Green PCR Kit, in a StepOnePlus Real-Time instrument (Applied Biosystem Life Technologies, Carlsbad, CA, USA). Normalization was done on specific miRNAs included in the array (RNA U6, SNORD61, SNORD68, SNORD72, SNORD95, SNORD96A). Analysis of the array results was performed using the specific miscript miRNA PCR array Data Analysis Excel spreadsheet. The threshold was set at ±2-fold to highlight/reveal significant expression regulation. The experiments were performed twice (n = 2).

### 2.4. pMirGLO Transfection and Dual-Luciferase Reporter Assay

To obtain the inhibition of mir-124-3p, we used the plasmid pmirGLO Dual-Luciferase miRNA target expression vector (Promega, Madison, WI, USA), which uses firefly luciferase, as the primary reporter to monitor miRNA regulation, and *Renilla* luciferase as an internal control reporter for normalization. For cloning, the plasmid was cut with the restriction enzymes SacI and SalI, present in the Multiple Cloning Site at the 3′ of firefly luciferase gene and ligated in frame with a double-stranded oligonucleotide containing the miRNA of interest in 5′-3′ direction. The primer sequences used to construct the double strand miRNA are shown in Appendix A. The ligation reaction was transformed into competent JM109 *Escherichia coli* cells (Promega, Madison, WI, USA) in liquid L-Broth (LB, Sigma-Aldrich, St. Louis, MO, USA) and in Petri dishes with Ampicillin (50 µg/mL). The recombinant colonies were selected by colony PCR (using forward and reverse primers external to the cloned fragment, respectively, Luc2 and LucR, shown in Appendix A), checked on agarose gel and subjected to electrophoresis. Integrity of inserted fragments was confirmed through sequencing (Bio-fab Research, Rome, Italy). The plasmid DNA was extracted using the PureYield Plasmid Miniprep System kit (Promega, Madison, WI, USA) and quantified by means of a bio-photometer D30 (Eppendorf, Hamburg, Germany). RAW264.7 cells were seeded in 96-well plates (2 × 10^4^ cells/wells) in α-MEM media supplemented with 10% FBS and left overnight to adhere. Then, cells were transfected with 0.05 μg luciferase reporters/well in medium without FBS using Lipofectamine 3000 (Invitrogen, Carlsbad, CA, USA). Six hours post transfection, OC differentiation was induced with RANKL treatment (50 ng/mL) and the cells incubated for further 24 h. Then, cells were analyzed for firefly luciferase activity, using the Dual-Glo luciferase assay system (Promega, Madison, WI, USA), according to the manufacturer’s instructions. Normalized firefly luciferase activity (Firefly/*Renilla* control) was compared to that of pmirGLO vector without insert. The instrument used was the Promega™ GloMax^®^ Plate Reader (Promega, Madison, WI, USA). The experiments were performed three times (each sample performed in triplicate). ** *p* < 0.01, *** *p* < 0.001 versus control.

### 2.5. MiRNA Transfection

RAW264.7 cells were plated in 96- (2 × 10^4^ cells/well) or 6-well (5 × 10^5^ cells/well) plates, cultured in α-MEM media supplemented with 10% FBS and left overnight to adhere. Then, cells were transfected with 50 nM mouse miR-124-mimic (miR-124m), which mimics endogenous mature miR-124 (Dharmacon, Lafayette, CO, USA), or negative control (NC) (Qiagen, Germantown, MD, USA), using RNAiMAX transfection reagent (Invitrogen, Carlsbad, CA, USA). Six hours post-transfection, OC differentiation was induced with RANKL treatment (50 ng/mL) and the cells incubated for further 24 h. Then, cells were harvested to perform further analyses. Experiments were repeated three times. ** *p* < 0.01, *** *p* < 0.001 versus mimic control.

### 2.6. Quantitative RT-PCR (qPCR) Analysis of mRNA and miRNA Expression

Quantification of mRNA gene expression was performed using a real-time PCR (StepOnePlus, Applied Biosystems, Waltham, MA, USA), according to the manufacturer’s manual, using a comparative threshold cycle method with SYBR Green master mix [30]. QPCR was carried out as previously reported [4] using the following primers (Qiagen): QT001676692 (NFATc1), QT00108815 (MMP9), QT00131012 (Acp5, TRAP), QT01047032 (DC-STAMP), QT02589489 (CTSK2), QT00197568 (RhoA), QT009399 (RANK), and QT01658692 (GAPDH). The threshold cycle (CT) values were calculated against the housekeeping gene Glyceraldehyde-3-Phosphate Dehydrogenase (GAPDH), whose expression was not affected by the experimental conditions. The primer accuracy was confirmed by the “melting curve” during the real-time PCR. At least two distinct biological samples were examined for each gene and treatment. Data are expressed as mean ± S.D. For mouse mir-124-3p, mir-144-3p, mir411-3p, mir-880-3p, and mir-295-3p, we used TaqMan MicroRNA Assay protocol (Thermofisher, Waltham, MA, USA), U6 snRNA as reference gene for normalization, and QPCR Probe master mix L-rox (2X) (Biotechrabbit GmbH, Berlin, Germany). The qPCR was carried out as follows: 1 cycle 2 min at 50 °C; 1 cycle denaturing at 95 °C for 10 min; 38 cycles, melting at 95 °C for 15 s; and annealing/extension at 60 °C for 60 s.

### 2.7. Western Blot Analysis

RAW264.7 cells were transfected with miR-124 mimic or All Star negative control (NC) siRNA and cultured as described above for 24 h with RANKL. Then, cells were lysed and total cellular proteins were extracted using RIPA buffer (Cell Signaling Inc., Beverly, MA, USA). Western blot was performed as previously reported [4]. Membranes were incubated overnight at 4 °C with the following antibodies: NFATc1 (Santa Cruz Biotechnology, Santa Cruz, CA, USA) at 1:1000 dilution; p-ERK1/2 and ERK1/2 (Cell Signaling Inc., Beverly, MA, USA) at 1:1000; and β-actin (Sigma Aldrich, St. Louis, MO, USA) at 1:5000. The secondary antibodies Alexa Fluor 680 Goat anti-Rabbit (1:2000) and Alexa Fluor 800 Rabbit anti-Mouse (1:5000) (Molecular Probes, Life Technologies, Carlsbad, CA, USA) were incubated for 1 h at room temperature. Protein visualization and quantization were carried out with the LI-COR Odyssey scanner and software (LI-COR Biosciences, Lincoln, NE, USA). Blots were imaged using an Odyssey Infrared Imaging System (LI-COR, Lincoln, NE, USA) according to the manufacturer’s instructions. Experiments were repeated two times.

### 2.8. DIANA-miRPath v3.0 Database, KEGG Enrichment Analyses, and Target Gene Prediction

We used the algorithm DNA Intelligent Analysis (DIANA) DIANA-miRPath v3.0 software (http://snf-515788.vm.okeanos.grnet.gr/ (accessed on 12 July 2021)) to analyze the differentially expressed miRNAs found in PCR arrays [31]. This database allows for the assessment of miRNAs regulatory roles and the identification of controlled pathways, supporting all analyses for the Kyoto Encyclopedia of Genes and Genomes (KEGG) and Gene Ontology (GO) databases, as well as providing prediction of the potential miRNA targets. In particular, here we used the KEGG pathway database and TargetScan v.7.2 (http://www.targetscan.org/vert_72/ (accessed on 28 June 2021)), Tar-base v.8 (http://carolina.imis.athena-innovation.gr/diana_tools/web/index.php?r=tarbasev8%2Findex (accessed on 28 June 2021)), and MicroT-CDS v.5.0 (http://diana.imis.athena-innovation.gr/DianaTools/index.php?r=microT_CDS/index (accessed on 28 June 2021)) tools, available within the DIANA database. Furthermore, the DIANA-miRPath v.3.0 software allows for reverse-search, i.e., identifying the KEGG pathways of interest and individually evaluating all the miRNAs that might regulate that pathway, looking for target genes that have or not have been experimentally validated.

### 2.9. Statistical Analysis

Data are expressed as mean ± S.D. of at least three independent experiments. Statistical significance between two groups was determined by a two-tailed Student’s *T* test. * *p* < 0.05 was considered to indicate a statistically significant difference.

## 3. Results

### 3.1. miRNAs Expression in RANKL-Treated RAW264.7 Cells

We have previously shown the timing and behavior of pre-OCs differentiation after RANKL stimulation of RAW264.7 cells over a 4-day period and at different levels, i.e., cell morphology, cytoskeletal organization, protein distribution, signaling pathways role, and OC-specific gene expression [4,5,29]. In particular, based on our previous studies on the expression of NFATc1 showing that both mRNA and protein levels increased already on the first day of RANKL stimulation, we established that this was the best RANKL stimulation time to perform miRNA expression analyses. Therefore, in order to identify and characterize new miRNAs involved in osteoclastogenesis, total RNA extracted from RAW264.7 cells, treated or not (undifferentiated cells) with RANKL for 24 h, was used to perform miRNA PCR array analyses. The differentially expressed miRNAs with *p*-value less than 0.05 and fold change higher or lower than ±2.0 (threshold) were screened. The resulting PCR array dataset shown in the heat-map profile (Figure 1B) was derived from the normalization of RANKL^+^ versus RANKL^−^ cells (hereafter referred to as wild-type cells). The expression of 38 miRNAs out 84 (the total number analyzed, see Figure 1A) varied more than two-fold (i.e., beyond the threshold value). In particular, RANKL treatment caused the upregulation in the expression of 10 miRNAs (red in the heat-map profile), which we will identify as Group 1 from here on, whereas the expression of the remaining 28 miRNAs (Group 2) was downregulated (green in the heat-map profile). PCR array data are reported in Table 1. The downregulation of miRNAs such as 29a-3p, 32-5p, 19b-3p, and 144-3p, observed after RANKL-treatment, suggested that their target genes *CtsK*, *OSCAR*, *Fos*, and *MITF*, respectively, could increase.

### 3.2. Expression of miRNAs in NFATc1-siRNA/RANKL-Treated Cells

To find those miRNAs potentially regulated by NFATc1 activation, we depleted NFATc1 mRNA in RANKL-treated cells by using a specific siRNA, which silenced the expression of both mRNA and protein, as previously shown by Russo et al. [29] and in Appendix A. The resulting PCR array dataset, shown in the heat-map profile (Figure 1C), resulted from normalization of RANKL^+^ cells transfected with NFATc1 siRNA versus those transfected with NC-siRNA (negative control) and showed that expression of 10 miRNAs out 84 varied more than two-fold. In particular, NFATc1-knockdown treatment caused the upregulation in the expression of 6 miRNAs, which we will identify as Group 3 from here on, whereas the expression of the remaining 4 miRNAs (Group 4) was downregulated. PCR array data are reported in Table 2.

The PCR array data from the two treatments (wild-type and NFATc1-knockdown) were set according to a Venn diagram (Figure 1D). Interestingly, among the miRNAs undergoing expression changes, four were common between the two treatments, i.e., miR-880-3p and miR-295-3p, whose expression levels varied significantly with opposite trend in the two treatments, and miR-488-3p and miR-29b-3p, which were upregulated in both treatments.

### 3.3. Validation of PCR Array Results

We next performed a validation of the PCR arrays data on a selection of miRNAs, namely, miR-124-3p, miR-144-3p, miR-411-5p, miR-880-3p, and miR-295-3p, by qPCR and compared the results obtained with PCR array data (Figure 2). Excluding miR-295-3p, whose expression could not be determined in any treatment, and miR411-5p, which was undetermined in NFATc1-depleted cells, the trend of qPCR data (red bars) is in good agreement with those of PCR array (blue bars), i.e., the up- or downregulation of the various miRNAs was confirmed, even if some values were below the established threshold (higher or lower than ±2). In particular, in wild-type cells, the order of magnitude for values of miR-144-3p and miR-411-5p expression was visibly different between the two analyses, although the trend was in the same direction. This is probably due to the diverse methods of analysis, in agreement with what was reported by Prokopec [32].

### 3.4. KEGG Enrichment Analysis of Differentially Expressed miRNAs

To analyze the differentially expressed miRNAs highlighted by the PCR arrays, we used the groupings previously made for the Venn diagram (see Figure 1), i.e., Group 1 (wild-type/upregulated), Group 2 (wild-type/downregulated), Group 3 (transfected/upregulated), and Group 4 (transfected/downregulated). For each group, we searched for predictions on potential miRNAs’ functional roles in biological pathways and targets, by using TargetScan v.7.2 or TarBase v.8 or MicroT-CDS v.5.0 within the DIANA-miRPath v3.0 database and the KEGG molecular pathway database. A total of 46, 66, 35, and 36 KEGG pathways were significantly (*p* < 0.05) associated with Groups 1, 2, 3, and 4, respectively (see Appendix A). In general, the majority of the dysregulated miRNAs were predicted to be involved in human diseases (mainly related to cancer), signal transduction (i.e., MAPK, ECM receptor interaction, etc.), cellular processes (i.e., regulation of actin cytoskeleton, focal adhesion, etc.), metabolism (i.e., fatty acid metabolism/biosynthesis, etc.), organismal systems (i.e., osteoclast differentiation, insulin signaling pathway), and to a small extent in genetic information processing (i.e., protein processing in endoplasmic reticulum). Three of the enriched KEGG pathways are shown in Table 3 (for a complete view see Appendix A).

MAP kinases pathway (mmu04010) is an important signal transduction pathway involved in the differentiation and maturation of OCs. Our previous bioinformatic analyses (with GO) of differentially expressed genes (DEG) in NFATc1-silenced cells highlighted the correlation between NFATc1 and MAPKs [29]. Recently, we demonstrated that long-lasting phospho-ERK correlates with OCs migration/fusion [4]. Here, we highlighted the dysregulated miRNAs in NFATc1-silenced cells.

MAP kinases pathway was significantly associated with miRNAs belonging to all the groups here identified (Table 3). In fact, collective annotations of the miRNAs in the database returned a set of 8 out of 10 miRNAs for Group 1 (Appendix A), 26 out of 28 miRNAs of Group 2 (Appendix A), 6 out of 6 miRNAs of Group 3 (Appendix A), and 4 out of 4 miRNAs of Group 4 (Appendix A), involved in the control of MAPK signaling. For a general view of the genes in the MAPK pathway as predicted targets of miRNAs belonging to Groups 1 and 2, see Appendix A.

MiRNAs that regulate JNK1, p38s (alpha, beta, gamma, and delta) and ERK-2 belong to Group 3 (Appendix A), and those that regulate MEKK1-2 and dual-specificity phosphatases (DUSPs) belong to Group 4 (Appendix A). Moreover, single annotation in the DIANA database, for miR-31-5p, miR-335-5p, miR-let-7i-5p, miR-29a-3p, miR-29b-3p, miR-141-3p, miR-124-3p, and miR-295-3p, also revealed significant correlation with MAPK pathway (data summarized in Appendix A).

Among the cellular processes, it seems very interesting to analyze “regulation of actin cytoskeleton (mmu04810)” since it is fundamental for the regulation of migration and cell fusion, which are crucial steps for the formation of mature multinucleated OCs. Regulation of actin cytoskeleton was significantly associated with miRNAs belonging to Groups 1, 2, and 4 (Table 3). In fact, collective annotations of the miRNAs in the database returned a set of 7 out of 10 miRNAs of Group 1 (Appendix A), 24 out of 28 miRNAs of Group 2 (Appendix A), and 4 out of 4 miRNAs of Group 4 (Appendix A), involved in the control of this pathway. Within this pathway, we found miRNAs targeting different genes (for a general view see Appendix A). In particular, some interesting target genes of Group 4 miRNAs are Ras Homolog Family Member A (RhoA), Rho-associated kinase 2 (ROCK2), various Integrins (Itg), and Actin (Act) (for a list of genes see Appendix A). Reverse search in the DIANA database returned these other targets: Protein Phosphatase 3 Catalytic Subunit Alpha Isozyme (PPP3CA), Protein Phosphatase 3 Catalytic Subunit Beta Isozyme (PPP3CB), Protein Phosphatase 3 Regulatory Subunit B Alpha (PPP3R1), Protein Phosphatase 3 Regulatory Subunit B Beta (PPP3R2), Calmodulin-2 (CALM-2), Calcium/Calmodulin Dependent Protein Kinase 4 (CAMK4), Calcium Modulating Ligand (CAMLG), and Fn1 (Fibronectin) (data summarized in Appendix A).

Focal adhesion (04510, within “cellular processes”) and ECM–receptor interaction (04512, within “signal transduction”) are pathways that seem very interesting to analyze, partly because we have recently evaluated the differentiation of OCs on different substrates [5]. Focal adhesion pathway was significantly associated with miRNAs belonging to all the groups (Table 3), while ECM–receptor interaction was associated only with miRNAs of the group 3 (Table 3). Collective annotations of the miRNAs in the database returned a set of 7 out of 10 miRNAs of Group 1 (Appendix A), 24 out of 28 miRNAs of Group 2 (Appendix A), 6 out of 6 miRNAs of Group 3 (Appendix A), and of 4 out of 4 miRNAs of Group 4 (Appendix A), involved in the control of focal adhesion pathway; while 4 out of 6 miRNAs of Group 3 (Appendix A) were involved in the control of ECM–receptor interaction. Within these pathways, we found miRNAs targeting common interesting genes, such as Collagen Type I Alpha 1 Chain (Col1A1), Collagen Type I Alpha 2 Chain (Col1A2), Collagen Type 4 Alpha 1 Chain (Col4A1), Collagen Type 4 Alpha 2 Chain (Col4A2), Collagen Type 5 Alpha 1 Chain (Col5A1), Collagen Type 5 Alpha 2 Chain (Col5A2), Fn (Fibronectin), and Integrins (Itg) (see Appendix A and data summarized in Appendix A).

Intriguingly, osteoclast differentiation (mmu04380) was a pathway significantly associated only with Groups 1 and 2 (Table 3). However, genes such as *MITF* and *RANKL* (TNFSF11) or *MITF*, *NFATc2*, and *TRAF6* may be regulated by 6 out of 10 miRNAs Group1 and 23 out of 28 miRNAs Group2, respectively (see Appendix A).

### 3.5. miR-124-3p Expression in RAW264.7 Cells

The expression of miR-124-3p has been reported to be downregulated in BMMs stimulated with RANKL [25]. On the contrary, our results showed that the endogenous expression levels of miR-124-3p did not change, both by PCR arrays and qPCR, in all analyzed samples, i.e., wild-type and transfected cells. Nevertheless, as this miRNA is considered very important for osteoclastogenesis, we wanted to deepen this aspect and try to understand its potential role on the expression of some OC hallmarks in our experimental system (RAW264.7 cells) by means of a dual luciferase reporter assay system. The miR-124-3p inhibitor (anti-miR-124) was cloned into pmirGLO vector (pmirGLO-anti-miR-124), and luciferase activity was estimated in RAW264.7 cells transfected with the construct. The logic of this experiment is to sequester endogenous miRNA-124 by binding it to the complementary sequence contained within pmirGLO-anti-miR-124. The construct that remains free (i.e., not bound to the endogenous miR-124) will be able to produce luciferase luminescence, while the sequestered one will not. As expected, transient transfection of pmirGLO-anti-miR-124 significantly reduced the relative luciferase activity compared to control (empty pMirGLO vector, named pmirGLO-WT), although only by approximately 37% ± 3. This result probably indicates the presence of a low amount of endogenous miR-124-3p in the cells treated with RANKL, which fails to saturate all the anti-miR-124 introduced by the transfection (Figure 3A). To verify this hypothesis, cells were co-transfected with pMirGLO-anti-miR-124 and miR-124-mimic oligo (miR-124m); indeed, in this situation, the luciferase activity was almost completely abolished (90%) (Figure 3A).

Furthermore, we evaluated the potential role of endogenous miR-124-3p during osteoclastogenesis by qPCR. First, RAW264.7 cells were transfected with pmiRGLO-WT and then stimulated with RANKL for 24 h to rule out any technical problems due to transfection with the empty vector (Figure 3B). RAW264.7 cells transfected with pmiRGLO-WT without RANKL (CTRL-R) served as control. As expected, the osteoclastogenesis started after RANKL stimulation since the expression levels of *CtsK*, *DC-STAMP*, *NFATc1*, *MMP9*, *RhoA*, and *Acp5/TRAP* increased significantly compared to pmiRGLO-WT transfected cells without RANKL (CTRL-R) (Figure 3B). A further increase in the expression levels of these genes was observed in cells transfected with pmiRGLO-anti-miR124 and stimulated with RANKL for 24 h, compared to the pmiRGLO-WT control with RANKL (CTRL + R) (Figure 3C), probably as a consequence of endogenous miR-124-3p subtraction. On the other hand, an excess of miR-124-3p obtained by transfecting cells with miR-124m (mimic) and treating them with RANKL for 24 h caused a significant decrease in the expression of *NFATc1* and *MITF* compared to the negative control with RANKL (CTRL + R) (Figure 3D). Moreover, the expression of *Acp5*, *MMP9*, and *RhoA* did not change, while *DC-STAMP* and *CtsK* slightly increased compared to the control with RANKL (Figure 3D). The augmented endogenous levels of transfected miR-124-3p mimic were confirmed by qPCR (data not shown). Intriguingly, transfection with miR-124m did not change the levels of both NFATc1 and p-ERK1/2 proteins (Figure 3E).

## 4. Discussion

The specific aim of this study was to evaluate the potential role of new miRNAs in osteoclastogenesis, with particular focus on those that were correlated with NFATc1 expression. Therefore, we hypothesized that by analyzing miRNAs differentially expressed in RANKL-stimulated cells, followed by the comparison between them and those differentially expressed in RANKL-stimulated/NFATc1-silenced cells, we could have highlighted miRNAs both involved in OC differentiation and related to the NFATc1 expression. Based on this hypothesis, using RAW264.7 cells as pre-OCs and miRNAs PCR array, we observed the upregulated expression of 10 miRNAs (Group 1) and the downregulation of another 28 miRNAs (Group 2) in RANKL-stimulated cells, while six miRNAs were upregulated (Group 3) and four miRNAs downregulated (Group 4) in RANKL-stimulated/NFATc1-silenced cells. In the following, we will focus our discussion only on those miRNAs that seem the most interesting, based on our results, both present and past, and in relation to the data present in literature.

Computational research for miR-29a-3p and miR-29b-3p has indicated them as possible regulators of all the kinases of the axis MKK3-MKK6-p-38s (α, β, γ, δ), although each of them can act on different target genes. Interestingly, these miRNAs changed their expression levels in both wild-type (−2.87 miR-29a; +3.23 miR-29b) and NFATc1-silenced (+2.78 miR-29b) cells. A recent study reported that the expression of miR-29 family members increased during the differentiation in OCs derived from both BMM and RAW264.7 cells, and miR-29 family knockdown decreased OC formation [33]. The authors identified six new miR-29 targets negatively regulated, involved in commitment of precursor cells to the OC lineage and in cell migration, including *Calcr* (calcitonin receptor), *Cdc42* (cell division control protein 42), and *Srgap2* (SLIT-ROBO Rho GTPase-activating protein 2), and found decreased expression of *NFATc1* with the knockdown of all members of the miR-29 family (sponge miRNAs). Apparently in disagreement, our results showed that miR-29a-3p was downregulated in RANKL-stimulated cells, i.e., when pre-OCs start to differentiate [29], whereas miR-29b-3p was upregulated both in RANKL-stimulated cells and RANKL-stimulated/NFATc1-silenced cells. Even if these two miRNAs were categorized in different groups, our KEGG pathways analyses predicted that they might be involved in almost the same cellular pathways, i.e., osteoclast differentiation (mmu04380), MAPK signaling pathway (mmu04010), regulation of actin cytoskeleton (mmu04810), and focal adhesion (mmu04510). Moreover, Rossi et al. [34] reported that over-expression of miR-29b in human CD14^+^ peripheral blood mononuclear cells reduced OC formation, together with the decreased expression of *MMP9*, *NFATc1*, and *CtsK*, as well as actin ring rearrangement impairment. Further studies are needed to clarify the role of this miRNA family in osteoclastogenesis.

In our PCR array, miR-141-3p was strongly upregulated (+6.83) in NFATc1-silenced cells (Group 3). Our computational prediction analysis suggested that ROCK1 and ROCK2 are miR-141-3p targets. It has been reported that overexpression of miR-141 could inhibit the RhoA/ROCK pathway in human pulmonary arterial smooth muscle cells (hPASMCs) [35]. Recently, we have shown that *RhoA* expression did not increase significantly in 24 h RANKL-stimulated cells [5], but it was strongly decreased in NFATc1-silenced cells [29], suggesting that RhoA/ROCK pathway is inhibited.

It has been reported that miR-23a-3p is highly expressed in exosomes from RANKL-induced RAW264.7 cells [36] and negatively regulates GSK3β expression and activation in RANKL-stimulated BMMs [37]. In our PCR array, expression of miR-23a-3p was significantly downregulated (−86,904.84) in RANKL-stimulated/NFATc1-silenced cells (group 4), suggesting that GSK3β activity was restored.

Two other miRNAs that are correlated with NFATc1 expression appeared worthy of attention, namely, miR-880-3p and miR-295-3p, which were downregulated in wild-type cells and upregulated in NFATc1-silenced cells, thus suggesting a possible correlation between them and NFATc1. To date, there are very few papers in the literature mentioning these two miRNAs, which report data related to embryonic stem cell development [38] and to tissue repair induced by mesenchymal stromal cells [39]. Therefore, it seems that these two miRNAs are considered two regulators of reprogramming cell fate. Here, computational prediction tools returned some possible targets for these miRNAs, i.e., TNF receptor-associated factor 6 (TRAF6); CALM-2, a calcium binding protein; GSK3β, a serine-threonine kinase; DYRK1 and 2, which possess both serine/threonine and tyrosine kinase activities; and DUSP2 and DUSP8, all of which somehow directly or indirectly related to NFATc1.

TRAF6 is an intracellular signaling protein that responds to RANKL stimulation of pre-OCs, triggering a cascade of events regulating a number of transcription factors, including NFATc1 [40]. To date, TRAF6 appears to be regulated by miR-146a and miR-125a, but by our computational prediction, TRAF6 could also be a target of miR-880-3p.

CALM-2 activates the phosphatase calcineurin (CN), which in turn dephosphorylates its downstream target NFATc1, leading to its nuclear accumulation and activation in pre-OCs [40]. On the other hand, GSK3β can phosphorylate NFATc1 and confine it to the cytoplasm in an inactive form.

DYRK1 and DYRK2 are the major representative members of nuclear and cytoplasmic DYRKs, respectively, and are direct kinase regulators of NFATc1 and NFATc2 [41]. In particular, DYRK1 likely phosphorylates nuclear NFAT, promoting its nuclear export, whereas DYRK2 functions as a “maintenance” kinase by phosphorylating NFAT in the cytoplasm [42]. Furthermore, DYRK1 is considered a negative feedback regulator of NFATc1 since NFATc1 stimulates its expression, but, in turn, DYRK1 attenuates that of NFATc1 with the aim of controlling bone physiology [8]. DYRK1B is a validated target of miR-880-3p [38], as also suggested by our computational prediction analyses. Nevertheless, future studies are needed to confirm miRNAs targets and DYRK1B activity in the OCs differentiation.

DUSP2 is another possible target for miR-880-3p and miR-295-3p as suggested by our computational prediction. DUSPs are a family of ten different phosphatases that dephosphorylate MAPKs, thus playing a key role in type, duration, and magnitude of the signaling in mammalian cells. In particular, DUSP2 has been reported to in vitro dephosphorylate ERK2 and p-38α [43]. However, DUSP2-deficient macrophages and mast cells in vivo showed increased phosphorylation of JNK, but an unexpected decrease in phosphorylation of ERK and p38 (probably occurring through a negative crosstalk with JNK), with, ultimately, a decreased expression of *NFAT*, *AP-1*, and *Elk1* [44]. Recently, we found that NFATc1-knockdown decreases ERK1/2 phosphorylation [4], and here we observed the upregulation of miR-880 and miR-295 in NFATc1-silenced cells. Further studies are needed to confirm DUSP2 as miR-880-3p and miR-295-3p target.

In the PCR array analyses, we strangely found no changes in miR-124-3p expression, although this miRNA was shown to be involved in osteoclastogenesis in mouse BMM cells. Indeed, Lee et al. [25] proposed a new role for miR-124 in the regulation of OC differentiation, acting as a negative regulator of *NFATc1*, *RhoA*, and *Rac1* expression. According to the authors, miR-124 was downregulated in response to 3 days of RANKL treatment in mouse BMMs. These results, in apparent conflict with ours, may depend on some factors, mainly the different cells used (BMMs vs. RAW264.7) and different RANKL treatment times (3 days vs. 24 h), which could provide different responses. On the other hand, our transient transfection of pmirGLO-anti-miR-124 by dual luciferase reporter assay suggested that the endogenous expression levels of miR-124 were very low in RANKL-stimulated cells but also showed that its inhibition actually increased the expression of *NFATc1* and other OC hallmarks analyzed. As a further confirmation, overexpression of miR-124 in RAW264.7 cells (by transient transfection with miR-124 mimic) suppressed RANKL-dependent *NFATc1* expression, as expected. In similar microarrays analyses, the screening of endogenous levels of miRNAs expression in RAW264.7 cells [45] or in BMMs [46] after treatment with RANKL for different time intervals (24–82 h or 5 days, respectively) did not show miR-124 among those differentially expressed with respect to untreated cells. Further studies are needed to clearly establish the role of miR-124 in osteoclastogenesis.

## 5. Conclusions

The results obtained are interesting, as, in addition to confirming the role of some miRNAs already known for their involvement in the process of mature OCs formation, they provide some new information on miRNAs that have not yet been correlated with osteoclastognenesis. We are aware that further studies are needed to clarify the fine regulation of OC differentiation operated by miRNAs, but certainly this study adds some elements for a full understanding of this complex scenario. Figure 4 partially summarizes our data and those in the literature, suggesting the involvement of relatively unknown miRNAs (i.e., miR-880-3p and miR-295-3p) in the indirect control of NFATc1 by regulating the phosphorylation/dephosphorylation of kinases, such as ERK and p38, through their action on specific enzymes (DYRKs, DUSPs, and GSK3), or through inhibition of proteins, such as TRAF6 and calmodulin. Overall, we predict that multiple levels of regulation controlling the stability of the NFATc1 protein might involve miR-880-3p and miR-295-3p, even though experimental studies are needed to verify this hypothesis.

In conclusion, miRNAs provide a rapid and reversible level of regulation of many cellular processes. Such post-transcriptional regulation may be very efficient in different fields, as in osteoclastogenesis, being an important potential therapeutic target for the treatment of OC-related bone disorders such as osteoporosis and osteopetrosis. The question of whether miRNAs could be future targets for drugs for therapeutic use in humans is still open.

## Figures and Tables

**Figure 1 biology-10-01080-f001:**
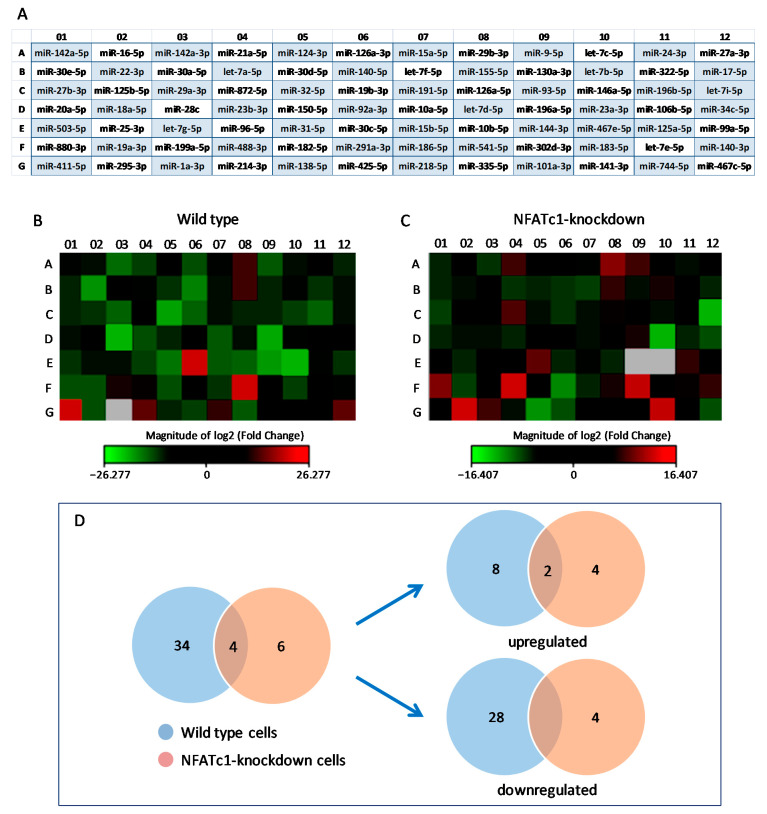
Differential expression profiling of miRNAs in RAW264.7 cells. (**A**) Heat-map legend showing the mouse miRNAs Profiler PCR array with all miRNAs analyzed. (**B**) Heat-map profile showing the resulting PCR array dataset derived from the normalization of RANKL^+^ versus RANKL^−^ cells (wild-type). (**C**) Heat-map profile showing the resulting PCR array dataset derived from the normalization of RANKL^+^ cells transfected with NFATc1 siRNA (NFATc1-knockdown) versus those transfected with negative control (NC) siRNA. Red squares, upregulated miRNAs; green squares, downregulated miRNAs; black squares, unchanged miRNAs; grey squares, technically unacceptable data. These latter were not considered in the analysis of our results. (**D**) PCR array data from the two treatments (wild-type and NFATc1-knockdown) were represented by Venn diagram.

**Figure 2 biology-10-01080-f002:**
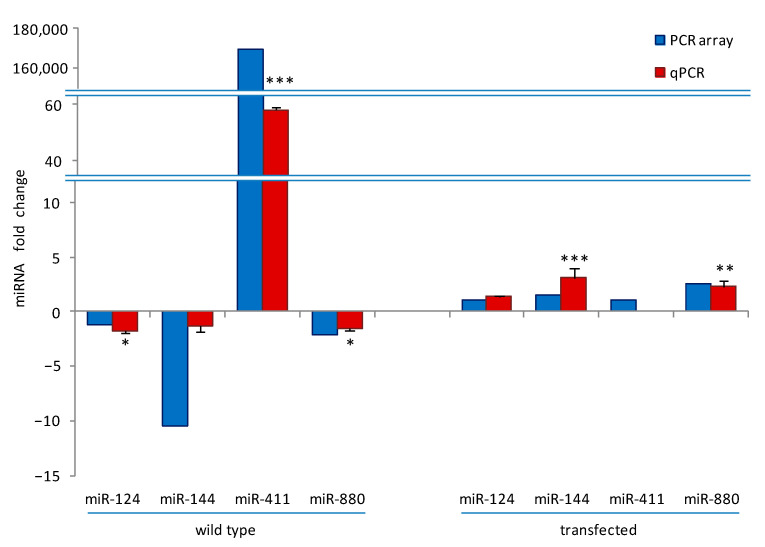
PCR arrays validation. QPCR data of the expression levels of miR-124-3p, miR-144-3p, miR-411-5p, and miR-880-3p in wild-type and transfected cells at 24h were compared with PCR array data to validate the trend of the expression changes. The PCR-array data reported here are the same as those reported in Table 1 and Table 2, except for miR-124 and miR-144, which are under the threshold (fold change ≥ 2). For qPCR, the miRNAs expression is presented as fold change of RANKL⁺ versus RANKL− cells in wild-type (control arbitrary set at 1) and in transfected RANKL⁺ cells as fold change of NFATc1 siRNA versus negative control (NC)-siRNA (control arbitrary set at 1). Each bar of qPCR data represents the mean ± SD of at least two independent experiments. * *p* < 0.05, ** *p* < 0.01, and *** *p* < 0.001, each agent alone versus control. U6 snRNA was used as a housekeeping miRNA for qPCR experiments.

**Figure 3 biology-10-01080-f003:**
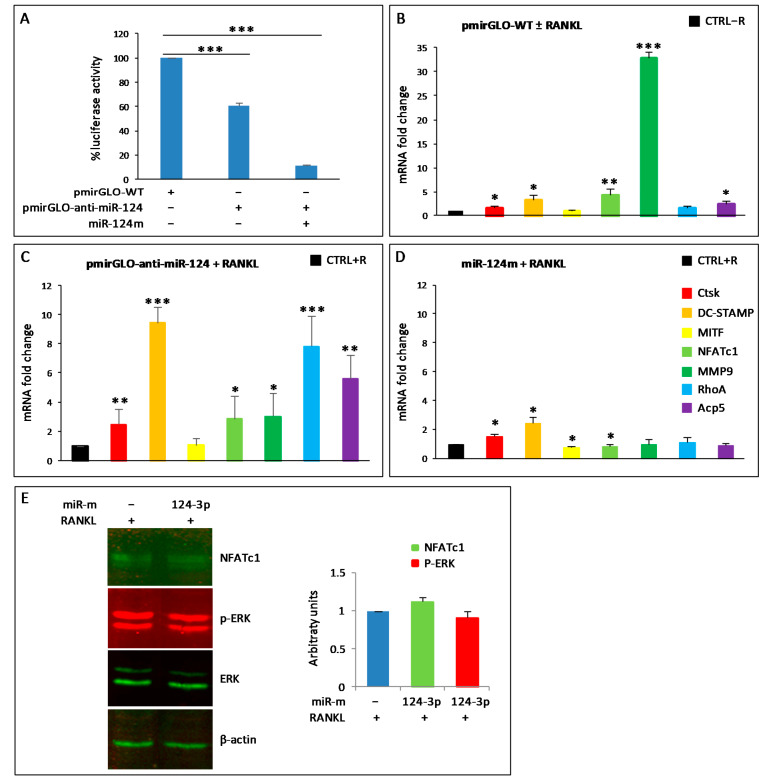
miR-124-3p expression in RAW264.7 cells. (**A**) RAW264.7 cells were transfected with control pmiRGLO-empty (WT) or pmiRGLO-anti-miR-124 without or with miRNA-124 mimic (miR-124m) and then RANKL-stimulated for 24 h. Luciferase assays were performed. The results shown are the means ± SD of five experiments (each of which was performed in triplicate). *** *p* < 0.001 each transfection versus control. (**B**–**D**) QPCR analyses were performed to evaluate the mRNA expression of *CtsK*, *DC-STAMP*, *MITF*, *NFATc1*, *MMP9*, *RhoA*, and *Acp*5/*TRAP* (see color legend in **D**). (**B**) RAW264.7 cells were transfected with control pmiRGLO-empty (pmiRGLO-WT), in the absence or presence of RANKL (± RANKL) for 24 h, and those without RANKL (CTRL − R) served as control. The results are shown as relative values of pmiRGLO-WT transfected cells with +RANKL vs. CTRL−R, arbitratily set at 1. (**C**) RANKL-stimulated cells (+RANKL) were transfected with pmirGLO-anti-miR-124, for 24 h. RANKL-stimulated cells transfected with control pmiRGLO-empty (pmiRGLO-WT) served as control (CTRL+). The results are shown as relative values of cells transfected with pmiRGLO-anti-miR-124 vs. those transfected with pmiRGLO-WT arbitratily set at 1. (**D**) RANKL-stimulated cells (+RANKL) were transfected with miR-124 mimic (miR-124m) for 24 h. RANKL-stimulated cells transfected with negative control (CTRL + R) served as control. The results are shown as relative values of cells transfected with miR-124m vs. those transfected with CTRL + R arbitrarily set at 1. GAPDH was used as a housekeeping gene in the qPCRs shown in (**B**–**D**), and the results are the means ± SD of three experiments (each of which was performed in triplicate). * *p* < 0.05, ** *p* < 0.01, and *** *p* < 0.001, each agent alone versus control. (**E**) RANKL-stimulated cells (+RANKL) were transfected with negative control (−) or miR-124 mimic (124-3p) for 24 h. Western blots were performed to evaluate the levels of NFATc1, ERK1/2, and p-ERK1/2. β-actin was a loading control. The data shown are representative of two independent experiments with comparable outcomes. The histogram shows the protein levels for NFATc1, normalized using β-actin, and p-ERK1/2, normalized using ERK1/2 and β-actin, detected on the same membranes. Each bar represents the mean value (±SD) of two independent experiments and shows the fold changes with respect to the control arbitrarily set at 1.

**Figure 4 biology-10-01080-f004:**
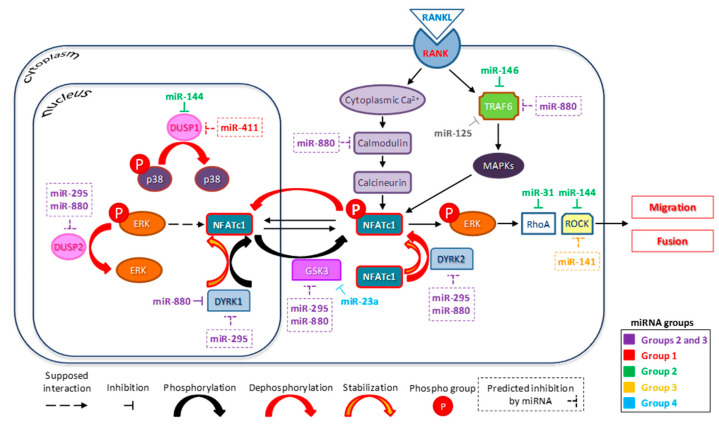
Diagram of the hypothetical mechanism controlling osteoclast differentiation involving miRNAs, ERK phosphorylation, and NFATc1 expression. MiRNAs differentially expressed in the PCR array are grouped as described in the text, i.e., Group 1 (untransfected/upregulated), Group 2 (untransfected/downregulated), Group 3 (transfected/upregulated), and Group 4 (transfected/downregulated). Solid arrows, known interactions (from literature or present results); dashed arrow, speculative interactions; −|, inhibition; and dashed rectangular, miRNAs predicted by our bioinformatic analyses.

**Table 1 biology-10-01080-t001:** Differentially expressed miRNAs in RANKL-stimulated RAW264.7 cells (fold change ≥ 2).

Upregulated (Group 1)		Downregulated (Group 2)	
miRNA	Fold Change	miRNA	Fold Change
miR-30c-5p	+690.085	miR-142a-3p	−3.285
miR-541-5p	+128.86	miR-21a-5p	−2.045
miR-411-5p	+169889.4	miR-126a-3p	−2.355
miR-214-3p	+3.45	miR-9-5p	−2.385
miR-467c-5p	+3.53	miR-22-3p	−8.17
miR-29b-3p	+3.23	miR-140-5p	−4.94
miR-199a-5p	+2.46	miR-29a-3p	−2.87
miR-488-3p	+2.02	miR-32-5p	−14.34
miR-218-5p	+2.83	miR-19b-3p	−2.905
miR-155-5p	+3.33	miR-146a-5p	−2.01
		miR-196b-5p	−2.92
		miR-28c	−3,252,400
		miR-23b-3p	−2.135
		miR-10a-5p	−2.7
		miR-196a-5p	−18.89
		miR-96-5p	−2.01
		miR-31-5p	−3.91
		miR-15b-5p	−2.395
		miR-10b-5p	−3.01
		miR-144-3p	−10.48
		miR-467e-5p	−15,364,809
		miR-880-3p	−2.13
		miR-19a-3p	−2.28
		miR-182-5p	−2.15
		miR-183-5p	−2.035
		miR-295-3p	−2.33
		miR-425-5p	−2.01
		miR-335-5p	−2.56
**10**		**28**	

In bold, the number of miRNAs for each group.

**Table 2 biology-10-01080-t002:** Differentially expressed miRNAs in RAW264.7 cells transfected with NFATc1-siRNA (fold change ≥ 2).

Upregulated (Group 3)		Downregulated (Group 4)	
miRNA	Fold Change	miRNA	Fold Change
miR-29b-3p	+2.78	miR-7i-5p	−10,459.28
miR-880-3p	+2.54	miR-23a-3p	−86,904.84
miR-488-3p	+16.21	miR-291a-3p	−3.65
miR-302d-3p	+7.36	miR-138-5p	−4.23
miR-295-3p	+18.97		
miR-141-3p	+6.83		
**6**		**4**	

In bold, the number of miRNAs for each group.

**Table 3 biology-10-01080-t003:** KEEG pathways analysis of the differentially expressed miRNAs. Differentially expressed miRNAs were grouped into the following groups (in bold): Group 1, wild-type/upregulated; Group 2, wild-type/downregulated; Group 3, transfected/upregulated; and Group 4, transfected/downregulated. Three enriched KEGG pathways are shown (in bold, light blue lines), i.e., signal transduction, organismal systems, and cellular process. *p*-value, threshold 0.05; # genes, number of genes regulated by miRNAs; # miRNAs, number of miRNAs involved in the KEGG pathway.

KEGG Pathway	*p*-Value	# Genes	# miRNAs
Signal Transduction			
**Group 1**			
MAPK signaling pathway (04010)	0.013507337	63	8
**Group 2**			
MAPK signaling pathway (04010)	0.000171124	105	26
**Group 3**			
ECM–receptor interaction (04512)	2.38994017785 × 10^−14^	4	4
MAPK signaling pathway (04010)	0.013613431	50	6
**Group 4**			
MAPK signaling pathway (04010)	2.61806457963 × 10^−5^	70	4
**Organismal systems**			
**Group 1**			
Osteoclast differentiation (04380)	0.02547096	36	6
**Group 2**			
Osteoclast differentiation (04380)	0.034318254	50	23
**Cellular processes**			
**Group 1**			
Regulation of actin cytoskeleton (04810)	0.000242141	64	7
Focal adhesion (04510)	0.022609292	53	7
**Group 2**			
Regulation of actin cytoskeleton (04810)	8.6120987 ×10^−6^	93	24
Focal adhesion (04510)	0.000412213	89	24
**Group 3**			
Focal adhesion (04510)	2.43074477409 ×10^−5^	53	6
**Group 4**			
Regulation of actin cytoskeleton (04810)	0.004173186	57	4
Focal adhesion (04510)	0.026505552	52	4

## Data Availability

Data are available upon request.

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
