# Peer review of "MiRNAs Expression Profiling in Raw264.7 Macrophages after Nfatc1-Knockdown Elucidates Potential Pathways Involved in Osteoclasts Differentiation"

_biology, 2021, doi:10.3390/biology10111080_

Round 1
Reviewer 1 Report
The article entitled “MiRNAs expression profiling in RAW264.7 macrophages after NFATc1-knockdown elucidates potential pathways involved in osteoclasts differentiation” by Russo et al., aimed to identify novel miRNAs that regulate osteoclastogenesis in RAW264.7 cells. Authors focused on bone resorption pathways where they conducted series of experiments using RAW264.7 cells.
Authors showed expression profile of miRNAs in RAW264.7 cells upon RANKL induction. Further they showed miRNA expression profiles in NFATc1 knockdown RAW264.7 cells. Finally authors performed in silico analysis of their results.
Overall this is very interesting study and to be considered for publication upon addressing below comments.
Comments:
- Key findings should be validated using an alternative model system.
- mRNA and protein expression of NFATc1 knockdown results should be presented.
- Font size of Figure-3 and 4 are too small to follow.
- Methods are too long.
Author Response
We want to thank the referees for their valuable suggestions thanks to which our manuscript has greatly improved. We hope that our changes are appreciated and that this revised version meets your expectations and satisfies reviewers.
Reviewer 1
Comments:
- Key findings should be validated using an alternative model system.
- A) The RAW264.7 macrophages used in our study are a widely acknowledged and suitable model system for the study of osteoclastogenesis and an extensive literature allows comparison with data of many authors. Actually, although an alternative model system might add some validation to our findings, we think there is no need to add another one here and, anyway, we would not be able to do so at the moment.
- mRNA and protein expression of NFATc1 knockdown results should be presented.
- A) the mRNA and protein expression of NFATc1 after its knockdown have already been shown in our previous paper Russo et al (2019), as quoted in the Results section 3.2. Nevertheless, as we usually check that siRNA transfection actually inhibits NFATc1 mRNA and protein expression, we prepared a supplementary figure (Fig. S1) with the requested data.
- Font size of Figure-3 and 4 are too small to follow.
- A) To overcome the font-size problem, Figures 3 and 4 have been removed and replaced with a Table 3, including only the pathways discussed in the Results/Discussion sections (as suggested also by the reviewer 3). All the pathways analyzed have been included in the Supplementary files, i.e. Table S2A, S2B, S3A and S3B.
- Methods are too long.
- A) We have shortened the Methods Section as requested.
Reviewer 2 Report
The current manuscript describes the expression of miRNAs in osteoclastic cell precursors, and the consequences of manipulating the expression of one miRNA on osteoclast gene expression. The authors identify novel miRNAs regulated by RANKL and by silencing the transcription factor NFATc1 in RANKL-treated cells. The study is relevant and could help understand better the mechanisms that lead to osteoclast differentiation and, potentially, dysregulation, leading to bone loss. It is, however, difficult to follow, and the nomenclature is not particularly helpful. In addition, there are some misconceptions and some of the interpretation of the data does not seem to be appropriate. Specific issues are detailed below.
- Simple summary: the statement indicating there are no successful therapies for osteoporosis is not correct, since treatments have been shown to stop or reverse bone loss successfully. Although better treatments might be developed, the ones currently used are far from being unsuccessful.
- The term “non-correlated” is not commonly used to describe non-targeting siRNA. A term like scramble, for example, is more appropriate.
- It would also help the reader if the cells that were not silenced are named wild type instead of untransfected.
- Materials and methods: It is not clear what the authors mean by “cells were suspended in…” in line 94? Were the cells seeded in these conditions?
- Cells not treated with RANKL should be named “undifferentiated” rather than resting, since they are probably proliferating in culture in the absence of RANKL.
- Another issue that needs clarification is how the “cut off value” was established, and what does it mean. Normally, if a measurement is below a cutoff, it should be considered undetectable, and that is not what it appears to mean in this manuscript.
- Authors seem to be raising conclusions from differences that do not reach significance in the results shown in figure 2, which is not acceptable.
- The rationale for studying miR-124, or at least why it was studied as a follow up of the miRNA analyses is not clear. If no changes were found by the authors, they should not rely in others to support the necessity of studying the effect of changes in the expression of the miRNA on osteoclastic gene levels.
- Authors should better explain how the experiments described under section 3.5 work, and how the luciferase expression is modified by miR-124 dysregulation.
- No indication of significant changes in NFATc1 levels are included in the graph for figure 5D, so it is not possible to conclude that miR-124m decreases the levels of the gene.
- It is not clear whether the levels of NFTc1 and pERK protein were corrected by the housekeeping gene. Since the authors also include the fold change in β-actin levels, and it is similar to the changes in pERK levels, one could conclude that pERK is not modified. On the other hand, the levels of NFATc1 are reduced when corrected by β-actin. Irrespective of whether this is the case, authors should provide replicas of the western blot, and the levels of proteins corrected by the housekeeping gene should be calculated and the mean and SD should be reported, with the proper statistical analysis. In addition, pERK levels should be corrected by total ERK.
Author Response
We want to thank the referees for their valuable suggestions thanks to which our manuscript has greatly improved. We hope that our changes are appreciated and that this revised version meets your expectations and satisfies reviewers.
Reviewer 2
The current manuscript describes the expression of miRNAs in osteoclastic cell precursors, and the consequences of manipulating the expression of one miRNA on osteoclast gene expression. The authors identify novel miRNAs regulated by RANKL and by silencing the transcription factor NFATc1 in RANKL-treated cells. The study is relevant and could help understand better the mechanisms that lead to osteoclast differentiation and, potentially, dysregulation, leading to bone loss. It is, however, difficult to follow, and the nomenclature is not particularly helpful. In addition, there are some misconceptions and some of the interpretation of the data does not seem to be appropriate. Specific issues are detailed below.
- A) we are aware that this manuscript is difficult to follow and therefore we have tried to clarify some points to make it easier to read.
- Simple summary: the statement indicating there are no successful therapies for osteoporosis is not correct, since treatments have been shown to stop or reverse bone loss successfully. Although better treatments might be developed, the ones currently used are far from being unsuccessful.
- A) As the reviewer rightly pointed out, we have deleted the sentence regarding osteoporosis therapy in the Simple summary section
- The term “non-correlated” is not commonly used to describe non-targeting siRNA. A term like scramble, for example, is more appropriate.
- A) As suggested by the reviewer we changed the “non-correlated” term, but we decided to use “Negative Control (NC)”, as in our experiments we used the AllStars negative control by Qiagen that is not a scramble siRNA. Indeed, a scrambled control usually involves a random rearrangement of the nucleotide sequence of the siRNA or shRNA under study. On the other hand, generally a non-targeting control is a siRNA/shRNA sequence designed not to target any known genes.
- It would also help the reader if the cells that were not silenced are named wild type instead of untransfected.
- A) As suggested by the reviewer, we changed the term “untransfected” with “wild type”.
- Materials and methods: It is not clear what the authors mean by “cells were suspended in…” in line 94? Were the cells seeded in these conditions?
- A) Actually, the cells were seeded in RPMI medium supplemented by FBS and antibiotics, then left to adhere and, 24h later, we changed the medium with α-MEM supplemented by FBS and antibiotics, along with RANKL to induce differentiation. To avoid confusion, we replaced “suspended” with “cultured” in line 94.
- Cells not treated with RANKL should be named “undifferentiated” rather than resting, since they are probably proliferating in culture in the absence of RANKL.
- A) As suggested by the reviewer, we replaced the term “resting” with “undifferentiated”.
- Another issue that needs clarification is how the “cut off value” was established, and what does it mean. Normally, if a measurement is below a cutoff, it should be considered undetectable, and that is not what it appears to mean in this manuscript.
- A) The cut-off value was arbitrarily set at ± 2 for the significantly differential expression of miRNAs, which means that values outside this range (obtained after normalization with the specific miRNAs included in the array) were considered variations of the expression, i.e. up or downregulation. On the contrary, the values included in this range (namely, under the threshold) do not necessarily mean that they are undetectable, as the StepOnePlus Real-Time instrument can record valid numbers, but rather indicate that there are no significant expression changes compared to controls, after normalization. In our PCR arrays, undetectable values are colored gray in the heat map, i.e. one miRNA for the wild type and 2 miRNAs for the transfected cells.
- Authors seem to be raising conclusions from differences that do not reach significance in the results shown in figure 2, which is not acceptable.
- A) We apologize, but we probably were unable to convey the validity and the reliability of the qPCR experiments performed to validate PCR array and, therefore, having highlighted the individual qPCR values, we involuntarily induced to focus the attention on the differences in the orders of magnitude of the values obtained by the two methods and obscured what they actually indicate. Indeed, it is known that profiling of mRNA abundance with different technologies/platforms may show good concordance, but with divergent sensitivity/time/price trade-off (Prokopek 2013). Therefore, since an acknowledged practice is to validate a subset of results using an alternate technology, what is expected is an agreement in the overall trend rather than good correspondence in the absolute values. Therefore, to clarify this point, we decided to replace the histograms for the four miRNAs in Fig. 2 with a new one showing a comparison between the results obtained by qPCR and those by PCR array. We believe that the validation of the trend of expression changes of the selected miRNAs is now quite clear. Considering that the qPCR experiments were performed at least two times, each bar represents the mean ±SD and the statistically significant variations to the relative control have been indicated by asterisks where possible.
Regarding the “conclusions raised from differences with no significance”, we would like to cite Amrhein et al. (Nature 567, 305-307, 2019), who discussed the need to be careful when statistics lead to denying differences that might instead be considered, albeit as apparently not significant .
- The rationale for studying miR-124, or at least why it was studied as a follow up of the miRNA analyses is not clear. If no changes were found by the authors, they should not rely in others to support the necessity of studying the effect of changes in the expression of the miRNA on osteoclastic gene levels.
- A) We may understand the reviewer’s query, but in our opinion it was interesting to pay attention particularly to this miRNA precisely because it is known as an important regulator of osteoclastogenesis, especially of NFATc1 that is the focus of our work, and the unexpected result of its unchanged expression caught our attention. Furthermore, to our knowledge, this is the first study on the potential role of miR-124-3p on the expression of some osteoclast hallmarks, other than NFATc1, in our experimental system (RAW264.7 cells). Thus, we first decided to validate the PCR-array results, which were confirmed by qPCR experiments, and then we tried to understand if the endogenous miRNA had a function even if its expression did not change in RANKL-stimulated cells and the results of the transient transfection with pmirGLO-anti-miR-124 proved that this was the case.
That said, we apologize to the reviewer but we have decided to leave these data, although we have tried to clarify our meanings throughout the text, in the Results and Discussion sections.
- Authors should better explain how the experiments described under section 3.5 work, and how the luciferase expression is modified by miR-124 dysregulation.
- A) As suggested by the reviewer, we tried to better explain this point changing Lines 490-494 as follows: “The logic of this experiment is to sequester endogenous miRNA-124 by binding it to the complementary sequence contained within pmirGLO-anti-miR-124. The construct that remains free (i.e. not bound to the endogenous miR-124) will be able to produce luciferase luminescence”, while the sequestered one will not.
- No indication of significant changes in NFATc1 levels are included in the graph for figure 5D, so it is not possible to conclude that miR-124m decreases the levels of the gene.
- A) The reviewer is right, from a data check it is evident that we forgot to indicate the significance (*p˃ 0.05), which was added as an asterisk in Figure 5D, which has now become Fig. 3D as a consequence of the transformation of Figs 3 and 4 in Table 3 (as requested by reviewer 1).
- It is not clear whether the levels of NFTc1 and pERK protein were corrected by the housekeeping gene. Since the authors also include the fold change in β-actin levels, and it is similar to the changes in pERK levels, one could conclude that pERK is not modified. On the other hand, the levels of NFATc1 are reduced when corrected by β-actin. Irrespective of whether this is the case, authors should provide replicas of the western blot, and the levels of proteins corrected by the housekeeping gene should be calculated and the mean and SD should be reported, with the proper statistical analysis. In addition, pERK levels should be corrected by total ERK.
- A) The reviewer is right since the numbers below the β-actin could be confusing. However, to include total ERK, we changed the images of WBs shown in Figure 5E (now Fig. 3E) with others coming from WB where NFATc1, ERK, p-ERK and β-actin were all analyzed in same membrane (see supplementary Fig. S2, which shows also two experiments with two independent biological samples). Furthermore, the new Figure 3E includes a histogram showing the protein levels of NFATc1, normalized with β-actin, and p-ERK, normalized with total ERK and/or β-actin. Each bar represents the mean value (± SD) of the two independent experiments shown in Supplementary Fig. S2.
Reviewer 3 Report
In the present study, the authors performed miRNA expression profiling of RANKL-stimulated RAW264.7 macrophages and NFATc1 silenced cells to identify the mechanism of osteoclast differentiation. They identified several miRNAs differentially expressed in the RANKL-stimulated RAW264.7 cells with or without NFATc1 silencing and deciphered the biological pathways and processes regulated by those miRNAs. The targets of miRNAs were also identified them. Several pathways regulated by NFATc1 was associated with osteoclast differentiation. Their approach was good but need substantial improvement. My specific comments are highlighted below:
- Line 121: The experiment should have been repeated three times.
- Figure 1: The heatmap is confusing. It is not clear which ones are not differentially expressed. From the figure, it seems like many of the miRNAs are up/downregulated. Authors need to change the scale or color coding of the heatmap. Also, the heatmap in Figure 1B & C should contain the miRNA names instead of having them in Figure 1A.
- Table 1: The expression fold change of some of the miRNAs are very high or low. How the normalization was performed? Is there any justification of >15 million fold change?
- Section 3.3: Authors must validate more miRNA expression for the reliability of the data.
- Line 275: Why did you validate miR-124-3p although it was below the cutoff?
- Lines 277-280: Although PCR array shows that miR-144-3p was downregulated in untransfected cells, it was not significant in qPCR analysis. Raise question about the reliability of PCR array data.
- Lines 283-285: Same goes for miR-880-3p which was not statistically significant in untransfected cells according to the qPCR analysis.
- Line 286: Why the expression of miR-295-3p was undetermined? Did you try to optimize the PCR condition and probe? Use different probe for this miRNA. Well, according to the Figure 2C, you have error bar for this undetermined miRNA expression. How can you have error bar if its undetermined? How many times it was repeated? and what was the CT values?
- Figure 2A: No error bars in the control group.
- Figure 2C: No error bars in the control group.
- Figure 3: This is too much information. Summarize the table. Alternatively, you can include the diseases as supplementary.
- Table 3: Please include this as supplementary.
- Lines 342-391: Why these pathways are important for the current study? Authors only talked about etc. pathways but didn't discuss their importance for the present study. It would be interesting to discuss the pathways regulated (may be uniquely) by NFATc1 since the title of the article focused on NFATc1.
- Line 474: Check the expression level of Junb. It may or may not be regulated in the same direction.
- The discussion is too long.
- The present study didn't actually studied any pathways. Authors should focus on some of the NFATc1 pathways related to differentiation.
Reviewer 4 Report
This manuscript describes miRNA expression profiling in murine osteoclastogenesis after NFATc1-knokdown using by siRNA. Although investigating miRNA expression profiling involved in NFATc1 expression is important, some points are insufficient. NFATc1-knokdown experiments using by siRNA have off-target effects. Therefore, you should use the cells from NFATc1 deficient mice. In addition, the experiment of over-expressing NFATc1 in RAW264.7 cells is needed. Thus, the results do not support the conclusion.
Author Response
We want to thank the referees for their valuable suggestions thanks to which our manuscript has greatly improved. We hope that our changes are appreciated and that this revised version meets your expectations and satisfies reviewers.
Reviewer 4
This manuscript describes miRNA expression profiling in murine osteoclastogenesis after NFATc1-knokdown using by siRNA. Although investigating miRNA expression profiling involved in NFATc1 expression is important, some points are insufficient.
- R) NFATc1-knokdown experiments using by siRNA have off-target effects. Therefore, you should use the cells from NFATc1 deficient mice.
- A) We disagree with the reviewer, as nowadays new siRNAs have been developed that show minimal off-target effects, such as those used in our work. Even though the use of cells from NFATc1 deficient mice would had new information, we do think that NFATc1-knokdown by siRNA is enough acknowledged considering the great amount of studies using this approach and the results obtained can lead to suitable conclusions. However, at the moment we are not able to perform new experiments using NFATc1 deficient mice, which of course might be taken into consideration for future studies.
R) In addition, the experiment of over-expressing NFATc1 in RAW264.7 cells is needed. Thus, the results do not support the conclusion.
- A) Even the gene over-expression usually provides interesting information, and in this study we considered the increased expression of NFATc1 stimulated by the addition of RANKL in the culture (see Russo et al 2019, 2020, cited in the manuscript ) as an endogenous over-expression, and compared undifferentiated (RANKL‐) vs RANKL+ cells and non-transfected (wild type) vs siRNA-NFATc1 transfected cells, all stimulated with RANKL. The results shown arise precisely from the comparison between the wild type (over-expression of endogenous NFATc1) and the silenced (inhibition of NFATc1) cells.
Round 2
Reviewer 3 Report
The authors addressed all the queries. I have no additional comment.
Author Response
Thanks again for the appropriate comments that have improved our manuscript.
Reviewer 4 Report
Although you revised the manuscript, the points I pointed out are not improved at all.
1) NFATc1-knokdown experiments using by siRNA have off-target effects. Therefore, the authors should use the cells from NFATc1 deficient mice.
You replied that siRNAs you used showed minimal off-target effects. Generally speaking, in order to reduce off-target effect to a minimum, researchers should use siRNAs as minimal concentration as possible (e.g. 1nM). However, you used 20nM siRNAs (Page3, Line 101). Thus, it is highly likely that NFATc1-knokdown experiments have off-target effects.
2) In addition, the experiment of over-expressing NFATc1 in RAW264.7 cells is needed.
You replied that addition of RANKL in the culture is substitute for the experiment of over-expressing NFATc1. However, as RANKL signaling cascade, many molecules (such as c-Fos, NFkB, MAPK and so on) are activated in up-stream of NFATc1. Therefore, addition of RANKL in the culture is not substitute for the experiment of over-expressing NFATc1.
Author Response
Although you revised the manuscript, the points I pointed out are not improved at all.
1) NFATc1-knokdown experiments using by siRNA have off-target effects. Therefore, the authors should use the cells from NFATc1 deficient mice.
You replied that siRNAs you used showed minimal off-target effects. Generally speaking, in order to reduce off-target effect to a minimum, researchers should use siRNAs as minimal concentration as possible (e.g. 1nM). However, you used 20nM siRNAs (Page3, Line 101). Thus, it is highly likely that NFATc1-knokdown experiments have off-target effects.
- A) The reviewer is partially right, as the off-target silencing effects may arise, but this could happen when the siRNA has partial complementarity in the seed region with unintended genes (Yuki Naito, BMC Bioinformatics 2009; Yuki Naito,Methods Mol Biol 2013). For this reason the Qiagen company had developed a new strategy to design the siRNAs, the “HP OnGuard siRNA Design”. Literally, from the Qiagen web site, “it is an upgrade to the HiPerformance siRNA Design Algorithm. It uses advances in the knowledge of the RNAi mechanism to reduce the risk of off-target effects and increase potency, ensuring successful RNAi. HP OnGuard siRNA Design incorporates an innovative neural-network algorithm, proprietary homology analysis, and exciting, newly added features. siRNAs are available at QIAGEN's GeneGlobeTM Web portal at qiagen.com/GeneGlobe. As well as siRNAs designed using HP OnGuard siRNA Design, matching RT-PCR assays, RNAi controls, and products for transfection and downstream analysis are available at GeneGlobe, covering the entire RNAi workflow. Discover more about HP OnGuard siRNA Design at www.qiagen.com/GeneGlobe!”. Moreover, we used FlexiTube Gene Solution (Qiagen), which is a gene-specific package of 4 siRNAs pre-selected for a target gene and the conditions used are those indicated by the manufacturer's manual. The mix of the siRNAs is 20 nM, but the concentration of each individual siRNA is 5 nM, which is an acceptable concentration. In addition, as reported by Qiagen on the web site: “FlexiTube GeneSolutions enable researchers to use multiple siRNAs for each target ensuring reliable results” and “FlexiTube siRNA and FlexiTube GeneSolutions are designed using innovative HP OnGuard siRNA Design and are available at QIAGEN's GeneGlobe Web portal. Design features include 3' UTR/seed region analysis, SNP avoidance, and interferon motif avoidance. siRNA design is reinforced using the results from QIAGEN's siRNA validation project in which thousands of siRNAs have been tested for effectiveness by real-time RT-PCR. Data from Affymetrix® GeneChip® analysis is used to minimize the potential for off-target effects”.
In general, the off target effects caused by siRNAs concentration are related to cytotoxicity. In this regard, we conducted experiments to control cell viability (MTS assay) after transfection, which is not affected compared to controls. Based on all these considerations, we do believe that the results obtained using siRNA NFATc1 can lead to suitable conclusions. However, we apologize but, as answered in our previous response to the reviewer, we are currently not able to perform new experiments using NFATc1-deficient mice, which of course might be taken into consideration for future studies.
2) In addition, the experiment of over-expressing NFATc1 in RAW264.7 cells is needed.
You replied that addition of RANKL in the culture is substitute for the experiment of over-expressing NFATc1. However, as RANKL signaling cascade, many molecules (such as c-Fos, NFkB, MAPK and so on) are activated in up-stream of NFATc1. Therefore, addition of RANKL in the culture is not substitute for the experiment of over-expressing NFATc1.
- A) We agree that the “addition of RANKL in the culture is no substitute for the experiment of over-expressing NFATc1”, and we didn’t intend it to be. Actually, since our main interest is the study of the early stages of osteoclastogenesis after the activation of the RANKL signaling cascade, we believed that it was more valuable to consider the endogenous “over-expression” of NFATc1, stimulated by the addition of RANKL in the culture, together with the “natural” activation of all the other molecules, up- and down-stream of NFATc1, and its silencing in RANKL-treated cells. We think that ectopic expression of NFATc1 without the RANKL stimulus would have deprived us of the overview of the RANK-RANKL signaling cascade, which is the beginning of osteoclastogenesis. We are aware that transcription factors like c-Fos and NF-kB and molecules like MAPKs are activated after RANKL, but it is precisely the overview that we wanted to have.
On the other hand, the ectopic expression of NFATc1 causes bone marrow-derived precursor cells to undergo osteoclast differentiation in the absence of RANKL (Takayanagi H et al. Dev Cell 2002; 3:889–901122) and can transform murine fibroblast in vitro (Lagunas and Clipstone, J. Cell. Biochem. 2009; 108:237-248). Even though the gene over-expression experiment certainly provides interesting and noteworthy information, we do believe that it was not suitable nor necessary in this study.
